Intragenomic polymorphisms among high-copy loci: a genus-wide study of nuclear ribosomal DNA in Asclepias (Apocynaceae)

Weitemier Kevin 1 kevin.weitemier@science.oregonstate.edu
Straub Shannon C.K. 2
Fishbein Mark 3
Liston Aaron 1
1 Department of Botany and Plant Pathology, Oregon State University , Corvallis, OR , USA
2 Department of Biology, Hobart and William Smith Colleges , Geneva, NY , USA
3 Department of Botany, Oklahoma State University , Stillwater, OK , USA
Lazo Gerard
Electronic publication date: 2015 Jan 6
Publication date: 2015
Volume: 3
Electronic Location ID: e718
Received 2014 Sep 23; Accepted 2014 Dec 11
Copyright: © 2015 Weitemier et al.
Copyright year: 2015
Copyright holder: Weitemier et al.
License: This is an open access article distributed under the terms of the Creative Commons Attribution License, which permits unrestricted use, distribution, reproduction and adaptation in any medium and for any purpose provided that it is properly attributed. For attribution, the original author(s), title, publication source (PeerJ) and either DOI or URL of the article must be cited.
License URL: https://creativecommons.org/licenses/by/4.0/

Keywords: Concerted evolution, Genome skimming, High-copy, Intragenomic polymorphism, Partial SNP (pSNP), Nuclear ribosomal DNA (nrDNA), Intra-individual site polymorphism, 2ISP, Asclepias, ITS

Funding: United States National Science Foundation Systematic Biology DEB 0919583 DEB 0919389 Funding for this work is supported by the United States National Science Foundation Systematic Biology program DEB 0919583 and DEB 0919389. The funders had no role in study design, data collection and analysis, decision to publish, or preparation of the manuscript.

==============================
Despite knowledge that concerted evolution of high-copy loci is often imperfect, studies that investigate the extent of intragenomic polymorphisms and comparisons across a large number of species are rarely made. We present a bioinformatic pipeline for characterizing polymorphisms within an individual among copies of a high-copy locus. Results are presented for nuclear ribosomal DNA (nrDNA) across the milkweed genus, Asclepias. The 18S-26S portion of the nrDNA cistron of Asclepias syriaca served as a reference for assembly of the region from 124 samples representing 90 species of Asclepias. Reads were mapped back to each individual’s consensus and at each position reads differing from the consensus were tallied using a custom perl script. Low frequency polymorphisms existed in all individuals (mean = 5.8%). Most nrDNA positions (91%) were polymorphic in at least one individual, with polymorphic sites being less frequent in subunit regions and loops. Highly polymorphic sites existed in each individual, with highest abundance in the “noncoding” ITS regions. Phylogenetic signal was present in the distribution of intragenomic polymorphisms across the genus. Intragenomic polymorphisms in nrDNA are common in Asclepias, being found at higher frequency than any other study to date. The high and variable frequency of polymorphisms across species highlights concerns that phylogenetic applications of nrDNA may be error-prone. The new analytical approach provided here is applicable to other taxa and other high-copy regions characterized by low coverage genome sequencing (genome skimming).

Introduction

With the advent of DNA sequencing technology to infer phylogenetic relationships, investigators began searching for genetic loci that were both phylogenetically informative and readily sequenced in most organisms. The use of nuclear ribosomal DNA (nrDNA) soon became a popular choice for phylogenetic inference (Hamby & Zimmer, 1988; Hillis & Dixon, 1991; Baldwin, 1992; Baldwin et al., 1995; Álvarez & Wendel, 2003). Nuclear ribosomal DNA offered several advantages over other loci: the combination of highly conserved and variable regions allowed phylogenetic inference across a broad range of evolutionary time scales; conserved regions allowed the use of “universal” PCR primers applicable to a wide range of taxa; the high copy number of nrDNA repeats allowed reliable amplification from lower quality DNA extractions; and the process of concerted evolution ensured that these copies were similar within individuals (Baldwin et al., 1995). The use of nrDNA, particularly the variable internal transcribed spacer (ITS) regions, became widespread, to the extent that many studies were based exclusively on ITS data (Álvarez & Wendel, 2003).

However, nrDNA loci have been shown to harbor limitations in their phylogenetic utility. Nuclear ribosomal DNA copies are assembled as tandem repeats at one or more loci in the genome, with each locus being known as an array. The number of repeats present within an array is labile, as is the number and location of arrays (Álvarez & Wendel, 2003). The process of nrDNA copy homogenization from homologous recombination or unequal crossing over is thought to occur much more frequently within than among arrays (Schlötterer & Tautz, 1994). Thus, differing nrDNA alleles may become fixed in different arrays within a genome, creating paralogy that, if unrecognized, may confound phylogenetic inference (Álvarez & Wendel, 2003; Song et al., 2012). Moreover, these events can create pseudogenes which, freed from selective pressures, may evolve through processes quite different from the functional loci and provide misleading evidence for between-individual genetic divergences if compared to functional copies (Buckler, Ippolito & Holtsford, 1997). These events may occur at a greater rate than inter-array homogenization via concerted evolution (Karvonen & Savolainen, 1993; Gernandt & Liston, 1999).

Due to the technical difficulty of systematically sequencing individual nrDNA loci because of their high copy number, studies characterizing the abundance and patterns of intragenomic nrDNA polymorphisms have been rare. Recently, studies utilizing whole-genome shotgun sequencing have begun to reveal levels of intragenomic polymorphism in Drosophila (Stage & Eickbush, 2007), nematodes (Bik et al., 2013), and fungi (Ganley & Kobayashi, 2007). However, these studies included a small number of species (12, 6, and 5, respectively) and did not attempt to place patterns of polymorphism in a phylogenetic context. Song et al. (2012) examined the ITS2 region of 178 plant species via pyrosequencing, finding nearly ubiquitous intragenomic variation, with most ITS2 copies within a genome represented by a few major variants. Other studies have used intragenomic nrDNA polymorphisms to identify populations of Arabidopsis (Simon et al., 2012) and infer intraspecific phylogenies of Saccharomyces (West et al., 2014). Studies of intragenomic nrDNA polymorphism patterns across many species within the same genus have not been performed in plants (but see Straub et al., 2012).

This study utilizes high throughput technology to survey many species and individuals in the angiosperm genus Asclepias (Apocynaceae) in order to characterize levels of intragenomic nrDNA polymorphism and place these within a phylogenetic context. The methods presented here are expanded from those we have previously developed as part of the Milkweed Genome Project (Straub et al., 2011; Straub et al., 2012), and generalized for use with a large number of taxa and any high-copy locus, such as those that may be obtained from a genome-skimming or Hyb-Seq study (Straub et al., 2012; Weitemier et al., 2014).

Methods

Sampling and sequencing

One hundred twenty-five individuals representing 90 Asclepias species and subspecies were sampled (Table 1) and sequencing libraries were produced as described in Straub et al. (2012). Two individuals of putatively hybrid origin were included: A. albicans × subulata and A. speciosa × syriaca. These individuals were collected from wild populations and identified as hybrids through expression of intermediate morphological characteristics (M Fishbein, 1996, 1998, unpublished data; see also Fishbein et al., 2011). Samples were multiplexed in approximately equimolar ratios, with up to 21 individuals per lane, and sequenced with 80 bp single-end reads on an Illumina GAIIx instrument (Illumina, San Diego California, USA). Asclepias subverticillata was multiplexed in a lane with 32 samples and sequenced with 101 bp paired-end reads on an Illumina HiSeq 2000 instrument, with reads analyzed as though they were single-end. One individual of A. syriaca was sequenced at higher coverage: this individual was sequenced in a single lane on an Illumina GAIIx with 40 bp single-end reads (Straub et al., 2011). To allow more efficient assembly downstream, read pools were filtered to remove plastid reads (using the custom script sort_fastq_v1.pl modified to retain Ns; Knaus, 2010).

Table 1 Polymorphic site abundance in Asclepias.

Polymorphic site abundance in Asclepias taxa.

Asclepias taxon	Voucher	Poly #	Poly %	High #	High %	SRA	
A. albicans S. Watson	Fishbein 3146 [WS]	174	3.12	16	0.29	SRS721451	
A. albicans	Fishbein 6463 [OKLA]	273	4.7	11	0.19	SRS721452	
A. alticola E. Fourn.	Steinmann 5243 [IEB]	668	11.45	31	0.53	SRS721453	
A. amplexicaulis Sm.	Lynch 12652 [OKLA]	248	4.25	25	0.43	SRS721454	
A. angustifolia Schweigg.	Reina 2004-1315 [ARIZ]	55	1.04	36	0.68	SRS721455	
A. angustifolia	Reina 2008-203 [OKLA]	169	2.99	26	0.46	SRS721456	
A. arenaria Torr.	Lynch 11495 [OKLA]	174	2.98	2	0.03	SRS721457	
A. asperula (Decne.) Woodson ssp. asperula	Lynch 12037 [OKLA]	342	5.87	7	0.12	SRS721458	
A. asperula ssp. asperula	Fishbein 6536 [OKLA]	391	6.73	24	0.41	SRS721459	
A. asperula ssp. capricornu (Woodson)
Woodson	Lynch 13314 [OKLA]	474	8.15	38	0.65	SRS721460	
A. asperula ssp. capricornu	Fishbein 6486 [OKLA]	347	5.95	6	0.1	SRS721461	
A. atroviolacea Woodson	Fishbein 3612 [ARIZ]	451	7.73	26	0.45	SRS721462	
A. auriculata Kunth	Lynch 1694 [OKLA]	513	8.85	33	0.57	SRS721463	
A. auriculata	Fishbein 5833 [OKLA]	378	6.54	39	0.68	SRS721464	
A. boliviensis E. Fourn.	Fishbein 6072 [OKLA]	875	15.08	111	1.91	SRS721465	
A. brachystephana Engelm. ex Torr.	Lynch 10642 [OKLA]	309	5.32	22	0.38	SRS721466	
A. californica Greene	Lynch 10779 [OKLA]	472	8.09	24	0.41	SRS721467	
A aff candida Vell.	Fishbein 6347 [OKLA]	245	4.2	14	0.24	SRS721448	
A. cinerea Walter	Fishbein 4793 [OKLA]	297	5.1	27	0.46	SRS721468	
A. circinalis (Decne.) Woodson	Webster 17186 [OKLA]	464	7.95	59	1.01	SRS721469	
A. connivens Baldwin ex Elliott	Lynch 12336 [OKLA]	394	6.75	16	0.27	SRS721470	
A. cordifolia (Benth.) Jeps.	Lynch 10942 [OKLA]	344	5.96	30	0.52	SRS721471	
A. cordifolia	Fishbein 5772 [OKLA]	308	5.35	13	0.23	SRS721472	
A. coulteri A. Gray	Ventura & Lopez 7986 [TEX]	471	8.09	27	0.46	SRS721473	
A. cryptoceras S. Watson ssp. cryptoceras	Fishbein 6504 [OKLA]	230	4	35	0.61	SRS721474	
A. cryptoceras ssp. davisii (Woodson)
Woodson	Fishbein 5723 [OKLA]	350	6	13	0.22	SRS721475	
A. curassavica L.	Zuloaga & Morrone 7087 [OKLA]	258	4.42	24	0.41	SRS721476	
A. cutleri Woodson	Fishbein 6511 [OKLA]	167	2.9	18	0.31	SRS721477	
A. cutleri	Fishbein 6500 [OKLA]	157	2.69	8	0.14	SRS721478	
A. emoryi (Greene) Vail ex Small	Carr 12032 [TEX]	124	2.29	44	0.81	SRS721479	
A. engelmanniana Woodson	Lynch 11224 [OKLA]	141	2.54	33	0.59	SRS721480	
A. engelmanniana	Lynch 11029 [OKLA]	192	3.3	7	0.12	SRS721482	
A. eriocarpa Benth.	Lynch 10923 [OKLA]	610	10.49	32	0.55	SRS721483	
A. eriocarpa	Lynch 10799 [OKLA]	492	8.43	17	0.29	SRS721481	
A. feayi Chapm. ex A. Gray	Fishbein 5586 [OKLA]	329	5.71	42	0.73	SRS721484	
A. fournieri Woodson	Fishbein 3660 [ARIZ]	556	9.58	47	0.81	SRS721530	
A. fournieri	Lynch 1655 [OKLA]	711	12.25	49	0.84	SRS721485	
A. glaucescens Kunth	Lynch 14142 [OKLA]	190	3.33	19	0.33	SRS721486	
A. glaucescens	Lynch 1623 [OKLA]	90	1.69	85	1.6	SRS721487	
A. glaucescens	Fishbein 5097 [OKLA]	215	3.72	18	0.31	SRS721488	
A. sp. nov. aff. glaucescens	Fishbein 3671 [ARIZ]	225	3.87	25	0.43	SRS721490	
A. hallii A. Gray	Lynch 11299 [OKLA]	600	10.36	54	0.93	SRS721449	
A. humistrata Walter	Fishbein 5596 [OKLA]	331	5.67	13	0.22	SRS721489	
A. hypoleuca (A. Gray) Woodson	Lynch 11374 [OKLA]	670	11.51	44	0.76	SRS721491	
A. incarnata L.	Lynch 12567 [OKLA]	434	7.45	28	0.48	SRS721492	
A. involucrata Engelm. ex Torr.	Lynch 12050 [OKLA]	326	5.65	35	0.61	SRS721494	
A. involucrata	Fishbein 6531 [OKLA]	217	3.72	13	0.22	SRS721493	
A. jaliscana B.L. Rob.	Fishbein 2493 [ARIZ]	140	2.97	74	1.57	SRS721495	
A. jaliscana	Fishbein 3657 [WS]	165	3	65	1.18	SRS721496	
A. jorgeana Fishbein & S.P. Lynch	Vásquez & Alvarez 4905 [IEB]	160	2.76	12	0.21	SRS721497	
A. lanceolata Walter	Fishbein 5605 [MISSA]	249	4.27	8	0.14	SRS721498	
A. lanuginosa Nutt.	Lynch 12661 [OKLA]	343	5.92	11	0.19	SRS721499	
A. latifolia (Torr.) Raf.	Lynch 11018 [OKLA]	236	4.05	11	0.19	SRS721500	
A. lemmonii A. Gray	Lynch 11453 [OKLA]	620	10.7	23	0.4	SRS721501	
A. leptopus I.M. Johnst.	Fishbein 6263 [OKLA]	282	4.84	9	0.15	SRS721502	
A. longifolia Michx.	Lynch 12447 [OKLA]	427	7.34	24	0.41	SRS721503	
A. lynchiana Fishbein	Venable & Becerra s.n. [ARIZ]	129	2.39	31	0.57	SRS721504	
A. macrosperma Eastw.	Gierisch 4191 [ARIZ]	76	1.42	30	0.56	SRS721505	
A. macrosperma	Fishbein 6518 [OKLA]	391	6.72	21	0.36	SRS721506	
A. macrotis Torr.	Lynch 11260 [OKLA]	364	6.27	11	0.19	SRS721507	
A. macrotis	Lynch 11263 [OKLA]	260	4.46	5	0.09	SRS721508	
A. masonii Woodson	Fishbein 3101 [OKLA]	151	2.59	7	0.12	SRS721509	
A. meadii Torr. ex A. Gray	Freeman 9106 [KANU]	208	3.62	20	0.35	SRS721510	
A. mellodora A. St.-Hil.	Zuloaga & Morrone 7168 [OKLA]	377	6.47	18	0.31	SRS721511	
A. mexicana Cav.	Fishbein 3009 [ARIZ]	186	3.22	18	0.31	SRS721513	
A. michauxii Decne.	Lynch 12316 [OKLA]	458	7.87	39	0.67	SRS721512	
A. notha W.D. Stevens	Lynch 14113 [OKLA]	375	6.45	44	0.76	SRS721515	
A. notha	Fishbein 5389 [OKLA]	249	4.31	41	0.71	SRS721514	
A. notha	Nee 32966 [NY]	432	7.47	17	0.29	SRS721516	
A. sp. nov. cf. notha	Fishbein 5816 [OKLA]	376	6.45	16	0.27	SRS721517	
A. nyctaginifolia A. Gray	Fishbein 6268 [OKLA]	109	1.92	23	0.4	SRS721518	
A. obovata Elliott	Lynch 11543 [OKLA]	708	12.75	87	1.57	SRS721450	
A. oenotheriodes Schltdl. & Cham.	Fishbein 5819 [OKLA]	230	3.95	15	0.26	SRS721519	
A. oenotheroides	Lynch 13339 [OKLA]	100	1.74	16	0.28	SRS721521	
A. oenotheroides	Lynch 11477 [OKLA]	134	2.3	7	0.12	SRS721523	
A. otarioides E. Fourn.	Bellsey 97-5 [ARIZ]	745	12.8	35	0.6	SRS721520	
A. otarioides	Lynch 1533 [OKLA]	659	11.32	20	0.34	SRS721522	
A. otarioides	Fishbein 5857 [OKLA]	697	11.95	48	0.82	SRS721524	
A. ovalifolia Decne.	Lynch 13546 [OKLA]	333	5.91	42	0.75	SRS721525	
A. ovata M. Martens & Galeotti	Laferrière 1478 [MO]	324	5.59	39	0.67	SRS721526	
A. pellucida E. Fourn.	Fishbein 5136 [OKLA]	439	7.53	9	0.15	SRS721527	
A. perennis Walter	Lynch 12408 [OKLA]	356	6.11	15	0.26	SRS721529	
A. pilgeriana Schltr. (“flava” in Fishbein et al., 2011)	Zuloaga & Morrone 7069 [OKLA]	458	7.85	23	0.39	SRS721528	
A. pratensis Benth.	Fishbein 5143 [OKLA]	410	7.11	29	0.5	SRS721531	
A. pratensis	Pérez 1850 [MO]	609	10.49	34	0.59	SRS721532	
A. prostrata W.H. Blackw.	Fishbein 2432 [ARIZ]	465	7.99	41	0.7	SRS721533	
A. purpurascens L.	Lynch 12847 [OKLA]	233	4.05	22	0.38	SRS721534	
A. purpurascens	Fishbein 5654 [MISSA]	216	3.73	7	0.12	SRS721535	
A. quadrifolia Jacq.	Webb s.n. [ARIZ]	116	2.33	41	0.82	SRS721536	
A. quadrifolia	Fishbein 6545 [OKLA]	204	3.55	31	0.54	SRS721537	
A. rosea Kunth	Lynch 1656 [OKLA]	747	12.86	33	0.57	SRS721538	
A. ruthiae Maguire	Riser 329 [WS]	31	0.61	23	0.45	SRS721539	
A. sanjuanensis K.D. Heil, J.M. Porter &
S.L. Welsh	Ellison s.n. [HPSU]	158	2.96	33	0.62	SRS721541	
A. sanjuanensis	Fishbein 6525 [OKLA]	237	4.11	38	0.66	SRS721540	
A. sanjuanensis	Riser 335 [WS]	68	1.18	26	0.45	SRS721542	
A. scaposa Vail	Fishbein 2951 [ARIZ]	500	8.61	35	0.6	SRS721543	
A. schaffneri A. Gray	Fishbein 5846 [OKLA]	327	5.65	36	0.62	SRS721545	
A. scheryi Woodson	Fishbein 5137 [OKLA]	586	10.05	15	0.26	SRS721549	
A. scheryi	Zamudio 5234 [MEXU]	272	4.88	35	0.63	SRS721544	
A. similis Hemsl.	Fishbein 3000 [ARIZ]	294	5.05	8	0.14	SRS721546	
A. similis	Fishbein 5148 [MISSA]	386	6.71	45	0.78	SRS721547	
A. solanoana Woodson	Lynch 10884 [OKLA]	882	15.13	35	0.6	SRS721548	
A. speciosa Torr.	Lynch 10981 [OKLA]	242	4.19	23	0.4	SRS721551	
A. aff. standleyi Woodson	Reina 98-579 [WS]	150	2.57	19	0.33	SRS721550	
A. subaphylla Woodson	Fishbein 3518 [WS]	182	3.31	23	0.42	SRS721552	
A. subaphylla	Lynch 1008 [OKLA]	383	6.86	77	1.38	SRS721566	
A. subulata Decne.	Fishbein 6434 [OKLA]	269	4.64	7	0.12	SRS721553	
A. subulata	Fishbein 6446 [OKLA]	370	6.36	18	0.31	SRS721554	
A. subulata × albicans	Fishbein 3142 [WS]	299	5.14	12	0.21	SRS721555	
A. subverticillata (A. Gray) Vail	Fishbein 2948 [ARIZ]	160	3.09	92	1.77	SRS721556	
A. syriaca L.	Lynch 11138 [OKLA]	204	3.51	11	0.19	SRS721557	
A. syriaca	Fishbein 4885 [OKLA]	298	5.1	7	0.12	SRP005621	
A. syriaca × speciosa	Fishbein 2810 [ARIZ]	161	2.89	37	0.66	SRS721559	
A. tomentosa Elliott	Fishbein 5608 [MISSA]	198	3.39	11	0.19	SRS721558	
A. tuberosa L. ssp. interior Woodson	Fishbein 2816 [ARIZ]	685	11.84	29	0.5	SRS721560	
A. tuberosa ssp. interior	Fishbein 4825 [MISSA]	297	5.1	31	0.53	SRS721562	
A. tuberosa ssp. rolfsii (Britton ex Vail) Woodson	Lynch 12526 [OKLA]	251	4.33	11	0.19	SRS721561	
A. uncialis Greene	Fishbein 6494 [OKLA]	282	4.86	19	0.33	SRS736934	
A. variegata L.	Lynch 12787 [OKLA]	375	6.5	29	0.5	SRS721563	
A. verticillata L.	Lynch 11102 [OKLA]	23	0.41	21	0.37	SRS721564	
A. vestita Hook. & Arn. ssp. parishii (Jeps.)
Woodson	Lynch 10735 [OKLA]	506	8.68	14	0.24	SRS721565	
A. viridis Walter	Lynch 12955 [OKLA]	261	4.55	40	0.7	SRS721567	
A. viridula Chapm.	Fishbein 4806 [MISSA]	425	7.37	46	0.8	SRS721568	
A. welshii N.H. Holmgren & P.K. Holmgren	Lynch 11369 [OKLA]	364	6.53	73	1.31	SRS721570	
A. woodsoniana Standl. & Steyerm.	D. A. Neil 242 [MO]	122	2.28	30	0.56	SRS721569	
Notes.

Voucher: Collector, collection #, [herbarium]; Poly #: Number of polymorphic positions; Poly %: Percentage of assembled positions that are polymorphic; High #, High %: Number and percentage of highly polymorphic positions; SRA: NCBI Sequence Read Archive accession number

An A. syriaca haploid genome size estimate of 420 Mbp and a nrDNA copy number estimate of 960 were used for estimates of sequencing depth and for comparisons with other organisms. These estimates are modified from Straub et al. (2011), where an incorrect estimate of the average A. syriaca 2C value led to a haploid genome size estimate of 820 Mbp and a nrDNA copy number estimate of 1,845. The current values are based on 2C estimates from Bai et al. (2012) and Bainard et al. (2012).

Polymorphism quantification

The method for determining polymorphisms present among nrDNA copies within an individual while retaining information about position homology across a group of distantly related individuals included four general steps, detailed below: (1) A sequence was selected to serve as a reference for the whole group; (2) A consensus sequence was obtained for each individual taxon and aligned against the group reference, allowing tailored read-mapping for each individual while associating positions along the individual consensus with their homologous positions in the group reference; (3) Reads for each individual were mapped onto that individual’s consensus sequence; (4) At each position the reads differing from the individual consensus were tallied.

Group reference

The nrDNA cistron of the high-coverage A. syriaca individual was previously assembled (Straub et al., 2011; GenBank JF312046). The nontranscribed spacer and external transcribed spacers from each end were removed due to the presence of internal repeats, and the more conserved 18S and 26S subunits used as the boundaries of the alignment. The resulting reference sequence contained 5,839 bp.

Individual consensus sequences

Read pools were examined prior to consensus assembly and reads that were exact duplicates were reduced to a single representative, retaining the highest average quality score (using the custom script fastq_collapse.py, available at https://github.com/listonlab). Sequences for each individual were constructed via reference-guided assembly, with A. syriaca as the reference, using Alignreads ver. 2.25 (Straub et al., 2011). Alignreads is a pipeline that includes the short-read assembler YASRA (Ratan, 2009), utilities from the MUMmer ver. 3.0 suite (Delcher et al., 2002; Kurtz et al., 2004), and custom scripts. Parameters were selected to ensure high identity of reads mapping to the nrDNA reference (95%), but allow the reconstructed sequence to differ from the reference. In addition to assembling the individual consensus sequence, Alignreads outputs a file associating each position in the group reference with those in the individual consensus.

Read mapping

Prior to mapping reads from an individual onto its consensus, reads with an average quality (Phred) score below 20 were removed, and bases in the remaining reads with a score below 20 were converted to Ns using FASTX-Toolkit ver. 0.0.13 (Gordon, 2008). Read mapping was performed with the program BWA ver. 0.5.7 (Li & Durbin, 2009), and output files processed with the SAMtools ver. 0.1.13 utilities (Li et al., 2009).

Reads were mapped onto the consensus sequence using the default mapping parameters in BWA. These allow up to 3 mismatches against the consensus in an 80 bp read and 4 mismatches in a 100 bp read, with long insertions or deletions excluded. In order to test the effect of relaxed mapping parameters on the abundance of polymorphisms detected, reads were mapped allowing 4 or 5 mismatches in an 80 or 100 bp read, respectively (the -n flag in bwa aln set to 0.015). The abundance of intragenomic indels was found by mapping reads using the default mismatch parameters, but allowing indels up to 5 bp long (bwa aln -e 4).

Polymorphism counting

A perl script was developed to tally the number of reads differing from the consensus at each position (polymorphic_read_counter_bwaPileup.pl ver. 3.03b, available at https://github.com/listonlab). For example, a base covered by 10 reads might have 7 reads with a G in that position and 3 with a C. In this case the consensus would have called a G at that position, with 30% of the reads differing. See File S1 for exact parameters and a pipeline of commands used.

Positions with 2% or more of reads differing from the consensus base were considered polymorphic. This cutoff is the same used by Straub et al. (2011) and comparable to that used by Nguyen et al. (2011) under a similar quality-filtering scheme. The control PhiX lane of the higher coverage A. syriaca individual was examined by Straub et al. (2011, Additional file 1 from that study) and found to have an error rate much less than 2%, indicating that the cutoff used here may be somewhat conservative. In addition to counting positions that were polymorphic, positions were recorded as “highly polymorphic” if 10% or more of the reads differed from the consensus.

In order to keep homologous bases aligned across individuals, only those positions that were present in the A. syriaca (group) reference were kept in the analysis (i.e., insertions relative to the reference were discarded). Note that deletions (relative to the reference) fixed within an individual are also not considered because zero reads called a base at that position.

RNA structure determination

The secondary structure of each subunit and spacer region was predicted for the A. syriaca reference from the minimum free energy structure found by the program RNAfold for the 18S, ITS1, and ITS2 regions and RNAcofold for the 5.8S + 26S regions (Lorenz et al., 2011). Program default model parameters were used (37 °C, unpaired bases can participate in up to one dangling end). Positions along the cistron were then categorized as either paired (stems) or unpaired (loops). Predicted structures are provided in File S2.

Polymorphisms within the cistron

The effects of cistron position (subunit or spacer) and secondary structure position (stem or loop) on the likelihood of a position being polymorphic or highly polymorphic were assessed in two ways. Differences in the likelihood that at least one individual was polymorphic at a position (e.g., positions coded as either polymorphic or invariant) were assessed via a two factor multiple logistic regression, as implemented in R ver. 3.1.0 using the MASS ver. 7.3.33 package (Venables & Ripley, 2002; R Core Team, 2014). Differences in the abundance of polymorphic individuals at a position were assessed using square-root transformed data with a two-way ANOVA and type III sum of squares for unbalanced design, as implemented using the car ver. 2.0.20 package (Fox & Weisberg, 2011). The ANOVA analysis is not reported for the highly polymorphic individuals because the data diverge substantially from assumptions of a normal distribution.

Phylogenetic context

A maximum likelihood estimate of phylogenetic relationships within Asclepias was produced by Fishbein et al. (2011, Fig. 2, TreeBase #27576). This tree was pruned to match the sampling in this study, and counts of polymorphic positions were recorded for each taxon. Taxa sampled in this study, but not present in Fishbein et al. (2011), were omitted from further analyses. Counts were averaged for taxa with multiple individuals in this study, but sampled only once in Fishbein et al. (2011). Ancestral states for the number of polymorphic positions in the rDNA cistron were reconstructed using squared-change parsimony in the Mesquite phylogenetic suite ver. 2.75 + build 573 (Maddison & Maddison, 2011; Maddison, Maddison & Midford, 2011).

The phylogenetic signal in the distribution of the number polymorphic positions was tested in two ways. In the first method, the total length of the tree (parsimony steps) in terms of changes in number of polymorphic positions was compared to a distribution of tree lengths created by 10,000 permutations of polymorphic positions across tips. In the second method, a likelihood ratio test (LRT) was performed between models where character evolution followed a Brownian motion model across the tree. In the first model, the parameter lambda (describing how well the phylogeny correctly predicts the covariance among taxa for a trait) was found that maximized the model’s likelihood (Pagel, 1999). This was compared to the likelihood found when lambda was held at zero, representing phylogenetic independence among species for that trait. Parsimony permutations were performed in Mesquite (Maddison & Maddison, 2011; Maddison, Maddison & Midford, 2011), and likelihood ratio tests were performed in R with the phytools ver. 0.4.05 and ape ver. 3.1.2 packages (Paradis, Claude & Strimmer, 2004; Revell, 2012; R Core Team, 2014).

To determine if any clades held intragenomic polymorphism frequencies that were significantly high or low relative to the rest of the phylogeny, polymorphism rates were simulated along the tree, and the true polymorphism counts compared to the distribution of simulated counts (Garland et al., 1993). Two tips in the tree separated by zero branch length (A. asperula ssp. asperula and ssp. capricornu) were collapsed, and the average of the polymorphism counts for the four sampled individuals of the species used for the new tip. A model of trait evolution under Brownian motion, using the lambda parameter estimated from the LRT above, was found from 10,000 random starting points using the fitContinuous function from the geiger ver. 2.0.3 R package (Harmon et al., 2008). This model was used to simulate polymorphism counts across the phylogeny 10,000 times, with lower and upper bounds of 0 and infinity, respectively, using the fastBM function in phytools ver. 0.4.31 (Revell, 2012). For each node, the ancestral state was estimated for the true data and the simulated data using squared change parsimony as implemented in Mesquite (Maddison & Maddison, 2011; Maddison, Maddison & Midford, 2011).

Results

Average coverage of the nrDNA region was ∼97× for all individuals (median = 88 ×; Sequence Read Archive PRJNA261980). Despite high overall coverage of the nrDNA region, not all positions of the cistron were assembled for all individuals; therefore results for polymorphic positions are presented both as counts and as percentages of sequenced bases. The A. syriaca reference had total genome coverage of ∼0.8× (Sequence Read Archive SRP005621).

The length of the reference A. syriaca nrDNA cistron was 5,839 bp. The consensus sequences for the nrDNA cistrons of other samples ranged from 5,815 to 5,865 bp, with over half of samples having lengths between 5,836 and 5,842 bp. Relative to A. syriaca, 87 samples include at least one inserted position in their consensus sequence, and 117 samples have at least one deleted position. However, because in general more positions were inserted than deleted, 71 samples have lengths greater than 5,839, and 41 samples have shorter lengths (Fig. S1).

Intragenomic polymorphism

All individuals were polymorphic at several positions homologous with the A. syriaca reference (Table 1). The number of polymorphic positions ranged from 23 (A. verticillata, 0.41% of sequenced bases) to 882 (A. solanoana, 15.13%), with a mean of 333 (5.77%, Fig. 1). A very high percentage (91%) of positions in the A. syriaca reference were polymorphic in at least one individual (Fig. 2A).

Figure 1 Polymorphic site frequency among species of Asclepias.

Histogram of polymorphic site frequency among species of Asclepias. Individuals contained from 0.4% to 15.1% polymorphic sites.

Figure 2 Polymorphic sites across the nrDNA cistron of Asclepias.

Number of individuals that are (A) polymorphic and (B) highly polymorphic at each position. Polymorphic positions are those with ≥2% of reads differing from the consensus; highly polymorphic positions are those with ≥10% differing reads. Subunit regions, white background; spacer regions, shaded background. Numbers in each region are the percentage of sites polymorphic or highly polymorphic in at least one individual.

Positions were significantly more likely to be polymorphic if they were in a spacer region (ITS1, ITS2; Fig. 2A) or stem (Fig. 3A). This is true both when considering the number of individuals polymorphic at a position (Table 2), and whether any sample was polymorphic at that position (Table 3A).

Figure 3 Polymorphism probability by region and structure.

Probability that at least one individual is (A) polymorphic or (B) highly polymorphic at a position that is either within a spacer (ITS1, ITS2) or subunit region (18S, 5.8S, 26S), and either paired (stems) or unpaired (loops). Error bars indicate 95% confidence intervals. Values derived from two-factor multiple logistic regressions (Table 3).

Table 2 ANOVA of the number of polymorphic individuals at nrDNA positions, by position type.

Two-way ANOVA of the number of polymorphic individuals at nrDNA positions categorized as either subunit (18S, 5.8S, 26S) or spacer regions (ITS1, ITS2), and as either paired (stems) or unpaired (loops). More individuals are likely to be polymorphic at sites that are in spacer regions over subunit regions, and that are paired over unpaired. Bold values indicate categories that significantly affect polymorphism abundance (P < 0.05).

Source of variation	Sum of squares	df	F value	P	
(Intercept)	8,310.6	1			
Paired	109.1	1	53.5887	<0.0001	
Subunit	16.4	1	8.0664	0.0045	
Paired* Subunit	0.4	1	0.1859	0.6663	
Residuals	11,878.9	5,835			
Notes.

df, degrees of freedom.

Table 3 Multiple logistic regression of the likelihood of nrDNA position polymorphism, by position type.

Two-factor multiple logistic regression of the likelihood of nrDNA positions being (A) polymorphic or (B) highly polymorphic in at least one individual. Positions are categorized as either within a subunit (18S, 5.8S, 26S) or spacer region (ITS1, ITS2), and as either paired (stem) or unpaired (loop). Odds ratios indicate whether a category decreases (<1) or increases (>1) the likelihood a position is polymorphic or highly polymorphic. The intercept represents paired, spacer positions. Categories that significantly affect polymorphism likelihood are indicated by italics (P < 0.1) or boldface (P < 0.05). Polymorphism probabilities for each category are presented in Fig. 3.

A: Polymorphic	
Source	Odds ratio	95% CI	Coefficient estimate	Std. error	Z-value	P	
(Intercept)			3.7136	0.3825			
Subunit	0.320	0.150–0.686	−1.1382	0.3881	−2.933	0.0034	
Unpaired	0.220	0.090–0.537	−1.5163	0.4561	−3.324	0.0009	
Subunit * Unpaired	2.488	0.998–6.206	0.9115	0.4663	1.955	0.0506	
B: Highly polymorphic	
(Intercept)			0.48551	0.1201			
Subunit	0.257	0.201–0.329	−1.35881	0.12564	−10.815	<0.0001	
Unpaired	0.719	0.494–1.047	−0.32964	0.19178	−1.719	0.0856	
Subunit * Unpaired	0.970	0.651–1.444	−0.03061	0.20315	−0.151	0.8802	
Notes.

CI confidence interval

Std. Error standard error

The number of highly polymorphic positions ranged from 2 (A. arenaria, 0.03%) to 111 (A. boliviensis, 1.91%) with a mean of 28 (0.50%). Positions highly polymorphic in more than 10 individuals were found in the 18S, ITS1, ITS2, and 26S regions (Fig. 2B). The most polymorphic position was 4,172 (using the A. syriaca reference), in the 26S region, which was highly polymorphic in 29 individuals. Highly polymorphic positions were dramatically less frequent in the subunit regions (18S = 19%, 5.8S = 26%, 26S = 31%) than in the spacer regions (ITS1 = 61%, ITS2 = 57%; Table 3B, Fig. 2B). Positions in secondary structure stems were moderately more likely to be highly polymorphic in at least one individual than loop positions (Table 3B, Fig. 3B).

Relaxed read mapping

Allowing more mismatches when mapping reads to their consensus nrDNA sequence increased the polymorphic sites counted within individuals by an average of 15.5% (Fig. S2). Two samples had no change in their polymorphism abundance, and two had a decrease (i.e., newly mapped reads at a previously polymorphic position matched the consensus, thereby dropping the polymorphic reads below 2%). The increase in polymorphism abundance under relaxed read mapping may indicate that the standard read mapping parameters are too conservative and that some truly polymorphic sites are excluded. However, because standard read mapping is more likely to exclude reads containing sequencing errors, and because there is a strong linear correlation (R2 = 0.97) between polymorphism abundance under the two mapping schemes, remaining analyses will only consider results from the standard read mapping.

Samples contained a mean of 7.6 and a median of 5 polymorphic indels when mapped reads were allowed to contain insertions or deletions of up to 5 bp relative to the individual consensus sequence. Eight individuals exhibited no polymorphic indels, including a sample of A. tuberosa ssp. rolfsii (Lynch 12526 [OKLA]). However, a sample of A. tuberosa ssp. interior (Fishbein 2816 [OKLA]) contained the most polymorphic indels at 51. Intragenomic indel abundance was positively correlated with SNP abundance (R2 = 0.35, Fig. S3). Due to the generally low level of intragenomic indel polymorphisms (78% of samples contained 10 or fewer), remaining analyses only consider results from intragenomic polymorphic SNPs. Polymorphic indels and SNP counts under relaxed mapping for each sample are provided in File S3.

Phylogenetic signal

The number of polymorphic base pair positions exhibited strong phylogenetic signal across Asclepias under both the permutation test (P < 0.0012) and the likelihood ratio test (estimated lambda = 0.51, P = 0.0067; Table 4; Fig. 4). This signal remained even after the ITS regions were removed from the dataset (permutation test P < 0.0200, lambda = 0.45, LRT P = 0.0112). Highly polymorphic base pair abundance, however, was not significantly influenced by phylogenetic history (Table 4; Fig. 5), even when considering only the subunit regions.

Figure 4 Ancestral state reconstruction of polymorphic site abundance.

Ancestral state reconstruction of the number of polymorphic positions in nrDNA in Asclepias obtained with squared-change parsimony. The tree topology is that pruned from Fig. 2 of Fishbein et al. (2011) with clades indicated by letters, following that study.

Figure 5 Ancestral state reconstruction of highly polymorphic site abundance.

Ancestral state reconstruction of the number of highly polymorphic positions in nrDNA in Asclepias obtained with squared-change parsimony. The tree topology is that pruned from Fig. 2 of Fishbein et al. (2011) with clades indicated by letters, following that study.

Table 4 Phylogenetic signal tests across Asclepias for the number of polymorphic or highly polymorphic sites.

Tests for phylogenetic signal across Asclepias for the number of polymorphic or highly polymorphic positions across the entire nrDNA cistron (Subunits + spacers) or just the subunits (Subunits only).

		Parsimony permutations	Lambda	logL	P	
Polymorphic sites	Subunits + spacers	<0.0012	0.51	−594.192−597.863	0.0067	
Subunits only	<0.0200	0.45	−587.547−590.765	0.0112	
Highly polymorphic sites	Subunits + spacers	>0.6050	<0.0001	−387.226−387.225	1	
Subunits only	>0.5300	<0.0001	−376.150−376.149	1	
Notes.

Parsimony permutations the proportion of permutations with a shorter tree length than the true data

Lambda the maximum likelihood estimate of lambda

logL the log-likelihood ratio of the unconstrained model including the estimated lambda over the constrained model with lambda = 0

P the probability of obtaining a likelihood ratio this small or smaller by chance alone

Of the 53 resolved clades in the phylogeny, none showed polymorphism values more extreme than expected under a Brownian motion model after correcting for multiple comparisons. The most extreme clade was that formed by A. hypoleuca and A. otarioides, which had an ancestral number of polymorphic positions exceeded by 79 of the 10,000 simulations (when excluding the spacer regions this node was exceeded by 60 simulations). The following most extreme nodes were those ancestral to A. rosea and A. lemmonii with more polymorphic positions than all but 130 (79) simulations, and ancestral to A. boliviensis and A. mellodora with more polymorphic positions than all but 246 (150) simulations. Under a Bonferroni correction a clade would require 9 or fewer simulations more extreme than the observed value (α = 0.05) to reject a hypothesis of no divergence from Brownian motion.

Despite the strong phylogenetic signal of polymorphism abundance across Asclepias, counts among samples within species (for those species with multiple samples) exhibited variability. Some samples of the same species had very similar polymorphism counts (e.g., the two A. jaliscana individuals contained 140 and 165 intragenomic polymorphisms), while others differed dramatically (e.g., A. macrosperma individuals contained 76 and 391).

Identification of mixed ancestry

The number of polymorphic sites for hybrid individuals, 299 for A. albicans × subulata and 161 for A. speciosa × syriaca, are less than the mean number of 333 polymorphic sites; and the number of highly polymorphic sites, 12 and 37, are less than or greater than the mean of 28. Of those positions that are highly po lymorphic, 4 of 37 in A. speciosa × syriaca have a minor allele frequency of 0.3 or higher, while none of the positions in A. albicans × subulata have a minor allele frequency above 0.2.

Discussion

Absolute counts of intragenomic polymorphisms among the copies of nrDNA in Asclepias (mean = 333 positions) were found to be much higher than levels reported for nematodes (<250; Bik et al., 2013), fungi (3–37; Ganley & Kobayashi, 2007), and Drosophila (3–18; Stage & Eickbush, 2007) when including all polymorphic positions. When considering only positions that are highly polymorphic, Asclepias exhibits slightly higher rates (mean = 28.4 positions) than fungi and Drosophila, but much lower rates than nematodes. However, these comparisons may be misleading: First, the number of nrDNA copies varies greatly between these taxa, estimated to range from about 50–180 in the fungal species, 200–250 in Drosophila melanogaster, 56–323 in the nematodes, and about 960 in Asclepias (Ganley & Kobayashi, 2007; Stage & Eickbush, 2007; Straub et al., 2011; Bik et al., 2013). Second, polymorphic base pair counts are confounded by differing criteria for scoring polymorphism (i.e., methods for excluding sequencing errors). The levels listed for the fungal species include both “high-confidence” and “low-confidence” polymorphisms, based primarily on sequence quality (Ganley & Kobayashi, 2007). The levels listed for Drosophila are polymorphisms present in ≥3% of loci (Stage & Eickbush, 2007). Bik et al. (2013) tallied read counts using a method similar to the method presented here, but called positions polymorphic when the count of differing reads exceeded what would be expected for a single copy locus. Third, sequencing depths of the fungal and Drosophila studies were much lower than those used here and with the nematodes (Ganley & Kobayashi, 2007; Stage & Eickbush, 2007; Bik et al., 2013). Nevertheless, given that Asclepias has absolute counts of polymorphic positions at least 33% higher than the other organisms studied, and that the sequencing depth was nearly two orders of magnitude greater in the nematodes than in Asclepias (6.3–10× per nrDNA copy in nematodes, vs. ∼0.1× in Asclepias; Bik et al., 2013), it is likely that Asclepias harbors greater rates of intragenomic polymorphism within the nrDNA cistron than the organisms studied to date.

Polymorphism patterns across the nrDNA cistron

Spacer regions (ITS1, ITS2) had higher frequencies of polymorphic positions than subunit regions (18S, 5.8S, 26S; Fig. 2). However, positions with low polymorphism frequencies are distributed much more evenly across the nrDNA cistron (Fig. 2A) than highly polymorphic positions, which show strong differentiation between the subunit and spacer regions (Fig. 2B). The lower frequency of highly polymorphic positions within the subunit regions suggests that these regions are under selection to remain homogenous within individual genomes. The lower difference in low polymorphism frequency between subunit and spacer regions suggests that this selection pressure is positively correlated with the proportion of nrDNA copies that differ from the majority. These findings contrast with those reported for nematodes (Bik et al., 2013), where the subunit regions had much higher levels of polymorphism abundance than the spacer regions.

Positions in stem regions were more likely to be polymorphic than loop positions (Fig. 3). This was strongly significant for all polymorphic positions (Tables 2 and 3A) and moderately significant for highly polymorphic positions (Table 3B). This would seem to contradict the hypothesis that stem sites in general should be more highly conserved in order to maintain a functional RNA secondary structure. Indeed, this finding agrees with those from Rzhetsky (1995), who not only found that trees estimated from stem regions contained longer branch lengths than those from loop regions, but that those stem sites least likely to affect secondary structure tended to be less conserved. Loop sites, on the other hand, contain a large proportion of the sites critical to ribosomal function (Rzhetsky, 1995), and may be under stronger stabilizing selection than stem sites.

Among the subunit regions, the 26S region harbored the highest frequency of highly polymorphic positions. This agrees with Stage & Eickbush (2007) who showed the Drosophila 28S region to have a higher mutation rate than the other subunit regions. However, in that study the 28S region also had a lower frequency of polymorphic base pairs. Stage & Eickbush (2007) hypothesize that this is due to the action of two retrotransposable elements found in many Drosophila 28S copies, with instances of aborted insertions causing the cell to repair the region using a nearby template and thereby homogenize the copies. They predict that levels of polymorphism will be higher in the 28S region of organisms lacking the retrotransposable elements, as seen here in the homologous angiosperm 26S region.

Within the subunit regions, highly polymorphic positions may be more common within expansion regions of the RNA gene (less conserved regions that tend to incorporate sequence insertions without affecting functionality; Clark et al., 1984). This is strongly implied by the recovery of clusters of highly polymorphic sites near A. syriaca positions 4,440 and 5,020, which are directly within the 26S expansion regions seven and eight, respectively (Kolosha & Fodor, 1990; Fig. 1). This agrees with results found in Drosophila (Stage & Eickbush, 2007). However, this may not be true for all highly polymorphic sites, as the high peak at position 4,172 is not within an expansion region.

Phylogenetic context

Mapping the number of polymorphic positions onto the phylogeny of Asclepias demonstrates strong and significant phylogenetic signal when counting all polymorphisms (Fig. 4), but this signal is not significant when only counting highly polymorphic positions (Fig. 5). While this study demonstrates phylogenetic signal in polymorphism abundance, it remains unknown whether abundance within a lineage is influenced by selection or purely neutral causes. Different demographic histories across clades could create this effect via neutral causes, while selection against organisms that retain too many variant nrDNA copies could be variable across the genus. Polymorphisms present in a high proportion of nrDNA copies may be uniformly selected against, while low polymorphism sites are tolerated at varying levels. This could explain the lack of phylogenetic signal at the highly polymorphic sites, and its presence when including all polymorphic sites.

These results have implications for phylogenetic inference based on nrDNA data. As previously cautioned, it cannot be assumed that all copies of nrDNA are identical within a genome. This is especially true for the spacer regions, but also for the subunit regions (Álvarez & Wendel, 2003). The discovery of phylogenetic signal in intragenomic polymorphism abundance demonstrates that those positions likely to lead to ambiguities are not distributed evenly across the phylogeny (Fig. 4). In addition to topology, variable polymorphism rates among lineages may affect the estimation of branch lengths on a tree. This may become especially problematic when nrDNA is used to date phylogenies. Recently developed methods of phylogenetic inference incorporating information about polymorphic positions may be able to alleviate difficulties in tree building caused by uneven levels of polymorphism abundance (Potts, Hedderson & Grimm, 2014).

Identification of mixed ancestry

Characterization of intragenomic nrDNA polymorphisms may allow for the identification of hybrid offspring between parents with differing nrDNA sequences (Zimmer, Jupe & Walbot, 1988), and may be able to provide an estimate of the number of generations since hybridization. This is especially true for early generation hybrids, as nrDNA homogenization can occur in a small number of generations (Kovarik et al., 2005), and mosaic Saccharomyces genomes have been identified based on intragenomic polymorphism abundance (West et al., 2014). Neither of the polymorphism profiles for the two wild-collected putative hybrid individuals in this study strongly indicate that they are early generation hybrids. This is in contrast to evidence from nuclear gene sequences that show heterozygosity consistent with inheritance of divergent alleles from the putative parents in the A. albicans × A. subulata hybrid (B Haack and M Fishbein, 2013, unpublished data). Detection of mixed ancestry may be hampered in this case by a lack of fixed differences between parental haplotypes. Consensus sequences for the A. albicans and A. subulata individuals in this study show no fixed differences, while one difference is found between the A. syriaca and A. speciosa consensus sequences. Unexpectedly, this position is monomorphic in the hybrid, while the positions with minor allele frequencies >0.3 are in positions not shown to differ between A. syriaca and A. speciosa. This likely indicates ancestry from an unsampled nrDNA haplotype.

Individuals of the same species with dramatically different polymorphism profiles may indicate the presence of cryptic diversity. Differing demographic histories between populations within a species may lead to individuals from those populations possessing different levels of intragenomic polymorphisms. As with the interspecific hybrids discussed above, early generation hybrids between populations within a species could also possess inflated levels of intragenomic polymorphisms.

Conclusions

Nuclear ribosomal DNA copies within individuals of Asclepias are not identical, with intragenomic polymorphisms present at a higher rate than reported for other organisms. Polymorphism frequencies across the genus vary by more than an order of magnitude and demonstrate strong phylogenetic signal. Stem positions of ribosomal subunits are more likely to be polymorphic than loop positions. Distribution of polymorphic sites across the nrDNA cistron are consistent with strong selection on nrDNA subunits, with polymorphic sites being more frequent in the spacer regions, and this difference being amplified for sites that are highly polymorphic. These results reinforce the need for caution when using nrDNA for phylogenetic inference, especially when using the spacer regions or for applications requiring the precise estimates of branch lengths or divergence times.

Supplemental Information

File S1 Command pipeline for tallying polymorphic positions

This file contains the commands used to tally polymorphic positions for each sample. It is formatted as a script that can be executed in a UNIX-like environment, but it will require individual modifications for each project.

Click here for additional data file.

File S2 Asclepias syriaca nrDNA cistron predicted secondary structures

These are the minimum free energy structures estimated from RNAfold (18S, ITS1, ITS2) or RNAcofold (5.8S + 26S). The first line of each entry gives the region name, the second line is the region’s sequence, the third line provides the region’s structure in dot-bracket notation followed by its minimum free energy (in kcal/mol). The 5.8S and 26S regions were analyzed together using RNAcofold. The sequences of each are separated from each other, in both the sequence and structure lines, by the ampersand (&) symbol.

Click here for additional data file.

Figure S1 Total consensus sequence insertions and deletions relative to Asclepias syriaca

Total number of inserted base pairs (Y-axis) and deleted base pairs (X-axis) of individual consensus sequences relative to A. syriaca. The size of the circle is relative to the number of individuals at that position. Points above the line are sequences longer than A. syriaca, points below the line are shorter.

Click here for additional data file.

Figure S2 Polymorphism abundance under relaxed vs. standard read mapping

The slope of the trend line is 1.211.

Click here for additional data file.

Figure S3 Intragenomic indels vs. SNPs

The slope of the trend line is 0.027.

Click here for additional data file.

File S3 Polymorphic site abundance in Asclepias, as comma-separated values

This dataset is identical to that presented in Table 1, with the addition of the polymorphic insertion/deletion abundance and polymorphic SNP abundance under relaxed read mapping. It is formatted as a table of comma-separated values.

Click here for additional data file.

The authors thank Richard Cronn for helpful comments and suggestions during the execution of this work. Expert laboratory assistance from Shakuntala Fathepure, Ben Haack, LaRinda Holland, Angela McDonnell, Laura Mealy, Nicole Nasholm, Matthew Parks, and Lauren Ziemian was a crucial contribution to this research. Kind thanks go to James Riser, who generously shared DNA extractions. Computer infrastructure was provided by the Center for Genome Research and Biocomputing at Oregon State University. Processing of sequences for the Sequence Read Archive was performed by Sanjuro Jogdeo.

Additional Information and Declarations

Competing Interests

Author Contributions

DNA Deposition

Data Deposition

The authors declare there are no competing interests.

Kevin Weitemier conceived and designed the experiments, performed the experiments, analyzed the data, wrote the paper, prepared figures and/or tables, reviewed drafts of the paper.

Shannon C.K. Straub performed the experiments, analyzed the data, reviewed drafts of the paper.

Mark Fishbein analyzed the data, contributed reagents/materials/analysis tools, reviewed drafts of the paper.

Aaron Liston conceived and designed the experiments, analyzed the data, contributed reagents/materials/analysis tools, reviewed drafts of the paper.

The following information was supplied regarding the deposition of DNA sequences:

Illumina reads for each samples have been placed in the NCBI Sequence Read Archive, and accession numbers are provided in Table 1.

The following information was supplied regarding the deposition of related data:

Custom scripts are available from GitHub:

www.github.com/listonlab/polymorphic_read_counter_bwaPileup

www.github.com/listonlab/fastq_collapse.

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
