# Peer review of "Intragenomic polymorphisms among high-copy loci: a genus-wide study of nuclear ribosomal DNA in Asclepias (Apocynaceae)"

_PeerJ, doi:10.7717/peerj.718_

## Round 0.1 · original submission · Minor Revisions

Your manuscript was reviewed by people very familiar with the subject and they appeared to have provided valuable criticisms which may add to the impact of the paper; please try to address the presented points as best as possible. In general the criticism was that the manuscript was well written and addressed a topic that would be of interest to the systematics community. In making these assessments it appears your depth of coverage criteria may help build assurances in phylogenetic studies, and yet have forewarned users of potential misreadings. The software provided and those used will help many undertake similar investigations which present similar challenges in resolving phylogeny. The contrasting differences seen between spacer and subunit regions in comparison with other studies may help provide a foundation for building assessment pipelines for analysis. As collections of genome surveys increase in depth and coverage, opportunities emerge to ask more detailed questions about genome organization. I approve this manuscript with minor revisions as recommended by the reviewers. Congratulations on your efforts.

Both reviewers had valuable input; addressing their suggestions would build the impact of this manuscript. Both reviewers pointed toward incorporating the NCBI-SRA references which would allow others to test your methods. As suggested the topic of insertions and deletions may have some merit, though it would seem these would be filtered out in the method presented here.

·

Basic reporting

Weitemeir, et al. present a thorough investigation of nrDNA sequence variation within the genomes of a broad sampling of species from the genus Asclepias. They developed a method to quantify intragenomic polymorphisms in nrDNA using whole-genome Illumina next-generation sequencing data and interpreted these results in the context of the Asclepias phylogeny.

The manuscript is well-written and clear. The analyses of general patterns of nrDNA variation, including which portions of ribosomal repeats vary, comparisons between species, and level of phylogenetic signal in amount of polymorphic repeats, are robust and interesting. I especially appreciate the mention of accommodating polymorphic positions in phylogenetic analysis, as there are many questions remaining regarding how we bridge the gap between traditional molecular systematics and next-gen methods. I recommend one general change (directly below).

I would like to see a mention of Song et al. (2012, citation below) at least in the introduction, as this study assessed ITS2 variation using pyrosequencing across a number of plant species (including 4 species from Cynanchum in Asclepiadaceae). While methodologically distinct in having sampled nrDNA alleles using PCR, citing this manuscript would provide valuable context to the analysis of levels of variation in Asclepiadaceae.

Song J, Shi L, Li D, Sun Y, Niu Y, et al. (2012) Extensive Pyrosequencing Reveals Frequent Intra-Genomic Variations of Internal Transcribed Spacer Regions of Nuclear Ribosomal DNA. PLoS ONE 7(8): e43971. doi:10.1371/journal.pone.0043971

Experimental design

The analytical pipeline and statistical testing for evaluating results are generally rigorous. I have one comment (directly below).

The maintenance of sequence homology relative to A. syriaca (pg 7 lines 127-130) is understandable as a way to more easily characterize and relate polymorphism between species. Is there a sensible way to briefly describe the levels of insertion and deletion among species? Indels are certainly important to accommodate when conducting phylogenetic analysis, and readers will likely be interested in variation in length for nrDNA copies. Additionally, is it possible that this methodological decision is slightly inflating (or deflating) the manner in which you're reporting polymorphism? I certainly don't expect you to completely resolve this issue (as it could certainly comprise a complete second manuscript!), but it is possibly a limitation of the study that shades how results are interpreted.

Validity of the findings

The manuscript indicates data collected for this study will be submitted to NCBI-SRA, and the scripts used to conduct the analyses are available in public repositories (GitHub) and as a supplemental file (which has very interpretable documentation and comments, although you have to read through the manuscript or code to find bwa and samtools are also dependencies). I expect these findings to be useful to many communities of researchers, including those interested in next-gen sequencing as well as molecular phylogenetics in plants. I have two comments about reporting results (directly below).

Table 1 indicates plant accessions from the same species possessed quite different proportions of polymorphic sites (I just glanced at A. angustifolia and A. cordifolia, which both exhibit one accession with twice the proportion of highly polymorphic sites). Given these values were averaged for the test of phylogenetic signal, I would like to see a mention in the results about intraspecific variation (this is also relevant to the discussion about identification of mixed ancestry, as well as the Song et al. paper I mentioned above).

The legend for Fig 2 does not differentiate between black and gray bars. I'm assuming these are just colored to differentiate between lines that are placed very closely together?

Additional comments

Thank you for doing a thorough job investigating a phenomenon that has plagued systematic botanists for years!

·

Basic reporting

The authors develop analyze polymorphisms within nuclear ribosomal DNA in the Asclepias complex, and compare the results to drosophila and nematodes. Overall, the paper is well written and adheres to the PeerJ policies.

Experimental design

Insertions and deletions are absent from the analysis, which can be used to support a majority of their conclusions. There may be a bias with how the reads were aligned to the reference using the alignment software, and needs to be addressed.

1) Important and absent from the paper is the cutoff criteria for read mapping to the generated reference nrDNA Line 107-112. This can reduce the possible number of polymoprhisms, especially at highly polymorphic regions (ITS), where a read can have a number of sites that differ from the reference. In supplemental data, there is no setting for the number of mismatches allowed with bwa “-n”. Default only allows for 0.04 in 80bp read ~ 3 mismatches. Authors need to ensure that increasing the number of mismatches does not dramatically increase their polymorphism counts.
2) Insertions and deletions (indels) are critical differences, should be within the data. Is there any reason these are excluded? Most algorithms including SAMTOOLS annotate indels differently from the site (e.g., insertion at position 10 would be position 10.1). Indels can have a larger influence on stem structure than base-substitutions. Seems that this is within the scope of the analysis.
3) Line 82. Step 1 is vague. What sequence? Reference sequence from the most closely related species available? Were the same reference sequence used for all consensus assemblies? This needs to be clarified if different references were used, and why they were.
4) Line 189 – SRA data left blank and in table 1 blank.

Validity of the findings

The general conclusions are justified assuming the issues in experimental design are not heavily biasing the results.

1) Conclusions 236-257 should be tempered as the distribution of intragenomic polymorphism is skewed upwards -- so estimates may not be that different from nematode. In addition, estimates in Dm and yeast are based off of low coverage WGS sequencing and may be an accurate survey of all variants. This should be explicitly stated.
2) A simple interpretation of the data is that Asclepias has nearly triple the number of estimated nrDNA copies than any of the compared organisms. Given that ~30-60 copies are essential in drosophila”, the nrDNA may be under less purifying selection and may have elevated levels of polymorphisms than other organisms with less copies.
3) Insertions and deletions can have a larger influence on stem structure than base-substitutions and should be at least examined in these regions.

Additional comments

Minor comments :
Line 31. Add “from homologous recombination or unequal crossing over”
Line 34. Homogenization can maintain function as well. Either remove this line or cite evidence that shows homogenization increases the probability of more psuedogenes.
Line 41/46/48. Define intragenomic to be within nrDNA.
Line 50. ‘The current study‘ to ‘This study’
Line 50. Awkward, change survey a large sample of species and individuals to - Multiple individuals belonging to different species in the angio sperm genus.

---

## Round 0.2 · accepted · Accept

The refinements made to the manuscript were appropriate and helped to resolve some of the reviewers' concerns. The reviewers were familiar with the message being developed and provided valuable comments. The added supplemental figures were a nice addition. The manuscript is in good shape and should move forward for publication; this will be passed forward for publication. You will be contacted by the editing staff for further refinements if necessary. I thank you for your contribution; your extended sample studies will surely serve as an example for, or perhaps a benchmark, for furthering studies in assessing species phylogeny. Congratulations.